# Food Security as Ethics and Social Responsibility: An Application of the Food Abundance Index in an Urban Setting

**DOI:** 10.3390/ijerph191610042

**Published:** 2022-08-15

**Authors:** Audrey J. Murrell, Ray Jones, Sam Rose, Alex Firestine, Joe Bute

**Affiliations:** 1David Berg Center for Ethics and Leadership, School of Business, University of Pittsburgh, Pittsburgh, PA 15260, USA; 2Food21 of Pennsylvania, Pittsburgh, PA 15139, USA; 3College of Business Administration, University of Pittsburgh, Pittsburgh, PA 15260, USA

**Keywords:** food insecurity, ethics, food systems, social responsibility, social inequalities

## Abstract

High levels of food insecurity signal the presence of disparities and inequities in local food access that have been shown to negatively impact the health and well-being of individuals and communities. Some argue that the lack of healthy, affordable and culturally relevant food within a community represents a troubling social and ethical concern for any society. The current research conducts an assessment of a specific community utilizing the framework outlined by the Food Abundance Index (FAI) scorecard. Combined with contemporary regional data on the demographics of the area, data revealed extremely low scores for both access and density dimensions. Our findings can help business, community and policymakers better understand and target evidence-based solutions to address the issue of food insecurity within this region.

## 1. Introduction

Food insecurity has been identified as a significant health and nutrition issue within the United States and globally [1]. High food insecurity has been associated with key health outcomes such as risk factors for children’s health and educational outcomes [2], long-term adverse consequence for overall health of children, maternal depression [3] and health status, mental health conditions [4], subsequent weight gain [5], poor sleep outcomes [6], chronic disease [7] and suicide ideation [8]. These data have led to research calling for immediate action to address food insecurity from a health, economic and social equality perspective. Some argue that the lack of healthy, affordable and culturally relevant food within a community represents an important social and ethical issue [9]. Food is considered a basic human right—as echoed by the United Nations, Declaration of Human Right, “people have a right to freedom from hunger, and everyone has a right to have access to adequate food” [10]. A right to food is a claim that is accepted and acknowledged by individuals and all of society [11]. This is because food, which is essential to all our lives, fulfills a basic biological need (along with water and oxygen) without which an individual would not survive or maintain their well-being [12,13].

Using this perspective, the current research agrees that satisfying this need for food is crucial for an individual and a community to maintain health, dignity and well-being. We see high food insecurity as a violation of one’s rights to a healthy and secure life and a denial of the opportunity to realize one’s full potential in society [14]. Consistent with the rights and common good approaches to ethics, access to food is part of society’s ethical and social responsibility as well as a necessary component of social sustainability [15]. In addition to an individual’s right to basic health and well-being, food insecurity may point to issues of disparity, discrimination or inequality [16]. For example, in the United States, food insecurity tends to be higher among households with incomes near or below the federal poverty line and black and Hispanic minority groups. Due to a variety of historical, political and social reasons, households with incomes near or below the poverty line are most likely to be minority households, single parent households, inner city or rural residents, elderly and people with disabilities [17]. The incidence of food insecurity among an economically and socially disadvantaged group raises concerns about ethics, social responsibility and economic equality [18].

### 1.1. Food Insecurity: An Ethics Analysis

A right to food access is a claim that is accepted and acknowledged by individuals and all of society. This is because food, which is essential to all our lives, fulfills a basic biological need without which an individual would not survive or maintain their well-being. Some theorized that ethics, public health and human rights are inextricably linked, and that people could not have rights and dignity if they could not maintain a healthy lifestyle [13]. This is similar to views that environmental, economic and social sustainability are linked as their collectively impact business and society. Clearly food insecurity with its negative consequences and its prevalence across community wide areas termed as “food deserts” are seen as a violation of ethics and social responsibility from a human rights perspective [19].

In addition to an individual’s right to basic health and well-being, food insecurity may point to issues of disparity, discrimination, or inequality [20]. For example, in the United States, food insecurity tends to be higher among households with incomes near or below the federal poverty line and black and Hispanic minority groups [21]. Due to a variety of historical, political and social reasons, households with incomes near or below the poverty line are most likely to be minority households, single parent households, inner city or rural residents, elderly and people with disabilities [22]. The incidence of food insecurity among an economically and socially disadvantaged group raises concerns about social justice, economic equality and discrimination within the current food system [23]. The inequitable distribution of benefits and burdens and the impact of these benefits and burdens that span all dimensions of social life including income, economic wealth, food, power, education, shelter, and health care brings to the forefront a number of basic ethical issues and concerns pertaining to food security [24].

### 1.2. Food Insecurity: A Social Responsibility Issue

Some would argue that high rates of food insecurity within the U.S. are the result of a fundamental conflict between a competitive market-driven food system and the overall needs of local communities and global societies [25]. A common good is often defined as certain or general conditions that are distributed equally to everyone’s advantage. The common good approach to ethics argues that social systems and environments on which we all depend must work in a manner that benefits all people [26]. But, current business practices are contrary to what one would argue are socially responsible business practices from a common goods perspective and has resulted in globally disparities, which have marginalized those at the bottom of the income distribution. For example, the current global food system consists of large-scale corporations whose strategy results in the marginalization of the small-scale producer thus reducing diversity in the number of firms involved across the food supply chain [27]. Other food retail practices like marketing unhealthy foods to kids and minority groups, higher pricing for organic and healthy foods, and a deliberate targeting of effluent and suburban consumers over those in low-income neighborhoods are all practices that are fail to recognize the interdependence between the health and welfare of communities and the competitiveness of businesses [28].

Clearly, food is one such common good that has been deemed important for the welfare of society. This is crucial for an individual to maintain his/her health and dignity and realize one’s full potential in society. Not only is food security a complex ethical issue but is also what is referred to as a “wicked problem”. A wicked problem is one that is tough to describe, has innumerable causes and require non-traditional approaches and processes in order to understand and solve complex wicked problems [29]. Food security also represents a wicked problem because it has a number of causes, has multiple perspectives that can be used to describe it, and doesn’t have a right answer for solving the consequences of this complex problem. Thus, any solution to the ethical and wicked problem of food insecurity, must involve multidimensional models and solutions in order to be effective [30]. Also, while the public sector has traditionally shouldered the responsibility for addressing issues related to hunger and food insecurity, increasingly studies have shown that private sector involvement and investment is critical to achieve comprehensive food security and build sustainable communities that co-create value for a wide array of stakeholders and that provides opportunities for economic and social security to all [31].

Therefore, the main objective of this paper is to describe a community-based study that utilized a data-driven tool that can help businesses, communities’ as well as policymakers identify areas of needs to target efforts to eliminate/reduce food insecurity. An understanding of the complex problem of food security, investigating the prevalence of food insecurity and identifying food insecure areas is an essential first step in executing transformational and socially responsible solutions to the complex issue of food insecurity. While awareness of the increase in the number of communities that are impacted by low food security has increased, existing tools to measure or track level of food security are complicated, limited in scope, expensive and not made available to local communities who need reliable information in order to address these negative conditions. Thus, a need for more reliable and accessible evaluation tools prevails [32].

Previous research has developed such as tool that is aimed at shedding additional light on the issue of food security called the “Food Abundance Index (FAI)” that integrates food deserts as one way of examining food security in geographic regions. The FAI, which uses a scorecard for assessing levels of food security across five key dimensions has been designed for broad use by individuals, communities, policy makers and businesses and reflects our unique approach to food security as a matter of both social responsibility and sustainability. The development and use of the FAI tool is based on studies on food deserts including geographic measures of food access in food deserts and prior efforts that examine retail food environments. Prior research on measures of food insecurity focus on key dimensions such as availability, affordability and quality of healthy nutritious foods and its effect on consumption behaviors [33]. These efforts are critical in order to provide an evidence-based assessment of food insecurity that can better direct solutions that take into account dimensions of food insecurity along with neighborhood demographics, impact on health and well-being and the impact of existing government interventions within targeted communities [34].

Clearly. solutions to address problems of food security need to be based on robust data about the extent of the problem and its consequences. While academics and scholarly researchers have been studying these issues for nearly two decades now, the peer reviewed academic research in which these studies and data is published is often not accessed by practitioners and policymakers [35]. On the other hand, reports on research by practitioners are often published in proprietary reports that come with costs not often within the reach of traditional academic researchers [36]. Our current research makes an initial attempt to bridge this gap by applying the Food Abundance Index to a local community as a test of its utility as a potential tool for understanding and driving business and social interventions to address food insecurity as an issue of ethics and social responsibility.

## 2. Research Setting

The Mid-Mon Valley region was selected for this particular research due to its unique history of rapid growth and targeted attention by public, private and community organizations concerned with the need for economic and environmental recovery. The decline of key industries such as steel had a substantial negative socio-economic impact on the region producing a systemic shock to the infrastructure of the region. In addition, the presence of revitalization projects specifically addressing food insecurity, most of which experienced significant volume increases during the COVID-19 pandemic, provided an indication that this region would provide an important test and application of the FAI framework.

The Mid-Mon Valley region was one of the largest and most complete steel making regions in the world for a period of nearly one hundred years [37]. Uniquely situated for access to coal, limestone and iron ore, it became one of the largest vertically integrated production centers in industrial history. At any one time every facet of steelmaking was going on continuously along the Monongahela River. Prominent manufacturing corporations had previously employed thousands of workers in the region throughout the 20th century. By the early 1970s only one significant corporation remained providing employment for up to 30,000 workers across all Mid-Mon Valley factories. As a result, the local economies and every other aspect of the community was directly tied to these facilities in the area which were operated by a single large manufacturer. Within the next decade, this corporation experienced over $561 million in losses which led to significant job and tax revenue loss. Layoffs and facility closures followed which results in severe loss in tax revenue, income, and population. During this time, unemployment rose to 20% and a significant number of residents lost their jobs due to the decline. Several regional initiatives and public policy efforts were created to address the economic, social and employment issues with the region [38].

The Financial Recovery Act of 1987 and the Mon Valley Initiative (MVI) sought to uplift economic activity in the region and have a significant impact on working towards a recovery in the region [39]. From a food security perspective, the Greater Pittsburgh Area Food Bank whose headquarters was located within the region was an important resource for food within these effected communities. These series of revitalization projects spark the potential for a resurgence of economic growth in the area for the future [40]. This provided an appropriate context for testing the FAI index together with regional data to examine this issue and identify targets areas for comprehensive solutions and additional regional investments.

For the size of the population, several of the ZIP codes within the Mid-Mon Valley have a significantly lower average income per household than the Allegheny County average. This region also has a higher minority population compared to the county as a whole. The region contained only one significant food production industry sector that was also one of the top ten employers in the region. According to the U.S. Bureau of Labor Statistics, the average food at home/away from home budget is approximately $8000 per year per household. However, the US census published that 12% of households are enrolled in Supplemental Nutrition Assistance Program (SNAP) benefits [41]. These SNAP benefits in the Mid-Mon Valley pay for approximately half of a family’s total food budget per year on average. This leaves these families with an additional expense of approximately $4000 per year, which can translate to close to 25% of their income. Lastly, the food industry sector as well as overall employment in the region, has been significantly negatively impacted by COVID-19 [42].

## 3. Methods

### The Food Abundance Index (FAI)

The original Food Abundance Index (FAI) was developed to measure the level of ‘food security’ and uses the presence of ‘food deserts’ as the key metric for assessing potential levels of food insecurity in a specific community [43]. The measure has been designed for broad use by individuals, communities, policymakers, businesses and corporations and reflects a unique approach to measuring food insecurity as a matter of both business responsibility and social responsibility. The FAI focuses on five dimensions that prior research show are key drivers of food insecurity.

*Access.* The access dimension is defined as the availability and ease of contact to healthy, nutritious and balanced food sources. When evaluating for access, the presence of food outlets selling at a minimum fresh produce, meat and dairy products are taken into consideration. These include mainstream grocery stores (large and small) and alternative food sources such as farmer markets, organic food outlets and local food source outlets like community supported agriculture, food cooperatives, farm stands, pick your own operation, etc. But the mere presence of such food sources is not enough, if people do not have direct access to the grocery store or food outlet. Since minority or economically disadvantaged communities often lack access to supermarkets within a short walking distance people have to rely on public transit to gain access to grocery stores. It was decided that to be classified as accessible, grocery stores must have a serviced bus stop within a five-minute or short walking distance within the designated study area.

*Diversity.* The diversity dimension first analyses the availability of multiple food sources selling healthy nutritious foods within the study area. Diversity of sources is important to look at because areas with high food insecurity usually lack healthful food sources like grocery stores, organic outlets, farmer markets, etc. which typically are outlets that have sell nutritious food items like fresh produce, meat, and dairy. This is also an issue of great concern because residents of food insecure areas would have to travel further to get essential food items or may have to depend on the one food outlet (in the absence of diversity) within the neighborhood offering unhealthy food products. Diversity also refers to the presence and availability of a variety of healthy and nutritious food items based on most relevant government sanctioned dietary and nutrition guidelines.

*Quality*. The duality dimension refers to the presence and availability of appropriately prepared, transported and preserved food that meets dietary needs of relevant community. At a minimum everyone should have access to adequately prepared and edible foods. A neighborhood without quality food forces residents to rely on fast food sources or convenience stores or travel outside of their neighborhood to achieve quality food for their family. This FAI dimension determines whether a food outlet sells fresh edible foods and no out-of-date or expired products from each food group of the relevant federal guidelines within the study area. A produce quality rating system is utilized to determine the freshness and appearance of products sold. One can also use existing consumer data on food quality from agencies like the health department to determine the quality of food available within the study area. Food stores can also play a significant role in promoting healthy buying, eating and consuming habits in their consumers. For example, the presence of healthy dietary intake promotion displays within the store like signage promoting low calorie foods, fruits and vegetables, organic or locally grown foods; labels near the food displays giving nutritional content of the food item; information on the food guide pyramid or tips for proper food storage and preservation creates a high-quality in-store food environment. Hence, healthy food promotion is also used as a part of the quality dimension.

*Density*. Density refers to the proportion of unhealthy food sources to healthy food sources present within the study area relative to health food sources. Food insecure communities typically have an imbalance between unhealthful and healthful food destinations and residents must rely on venues that sell highly processed foods and foods that are high in salt, fat and sugar. As a result, food insecure residents are more vulnerable to economic, social and physical consequences of living in areas with a dearth of outlets that sell nutritious foods. An ideal food environment should have a greater number of food outlets like grocery stores, produce vendors, organic and local food outlets like farmer markets, community supported agriculture, farm stands, etc. that provide a constant and reliable source for fresh produce, meat/poultry and dairy items as compared to convenience and fast-food stores.

*Affordability.* The final element of ‘affordability’ refers to the concentration and availability of healthy and nutritious food sources given the income and purchasing power of residents within the relevant geographic location. Firstly, affordability can be examined by the availability of healthy and nutritious foods at costs less than or equal to the national average cost of purchasing a ‘standard market basket’ of food items. The assessment uses the food list from the relevant federal guidelines, which provides a list of foods items plus condiments needed to prepare a week’s worth of food for an individual or a family. Store surveys can also be conducted to collect information on the lowest price at which food included in the market basket is sold for in the study area. Affordability can then be determined by comparing the cost of purchasing the complete market basket within the study area to the national average cost of purchasing based on the federal or government guidelines for the relevant assessment area.

The FAI is designed to provide a multi-dimensional tool for measuring the level of food security and especially whether a ‘food desert’ may exist within a specific community or neighborhood area. It attempts to combine the strengths of existing measures of food access and availability and examines food security across the multiple dimensions of access, diversity, quality, density and affordability. The FAI uses a scorecard approach similar to the LEED certification (Leadership in Energy and Environmental Design) for communities that awards points for actions across the five dimensions that enhance a community’s level of food security [44]. The scorecard approach allows individuals, communities and organizations to measure for levels of food security present in a geographic region, identify where problems or gaps may exist and track changes or progress across multiple dimensions [45].

For the current research. an updated version of the original Food Abundance Index (FAI) was used as the framework to identify and aggregate data directly from multiple public, research and community-based data sources. Our purpose was to apply the revised FAI tool as a framework for pooling relevant data and to combine, share and use these data resources to make decisions pertaining to local and regional food related policies and program efforts. This approach provides a better reflection of the key factors within each community related to the dimensions of the FAI that can provide a more detailed assessment of the level of food insecurity. There is a clear need for this type of aggregate data is in order to address potential gaps related to measuring food insecurity that exist with single item or single source data. With the current research, we used the framework of the FAI to select, collect and then aggregate existing data within current and publicly available data sets. To assess the Mon Valley region, the revised FAI tool was applied via a digital platform that included an interactive GIS map database that displays the region’s food outlets, which is updated monthly by the Western Pennsylvania Regional Data Center (http://www.wprdc.org/) (last accessed on 5 August 2022).

The enhanced FAI design provides numerous additional benefits. The sheer cost and contributed effort to conduct and maintain this program have been significantly reduced from the original assessment tool both by innovations in the aggregation of existing sources of data that can be updated more rapidly as conditions change and new information becomes available. By centralizing data from multiple sets in one application, and incorporating existing public and proprietary data sources many of the gaps in previous approaches can be addressed. Thus, the FAI tool combines pooling, sharing and using existing sources of data in order to use the current scorecard to collect, analyze, and employ these data to provide a more comprehensive examining of food insecurity pertaining to local and regional communities.

## 4. Results

Our results using the aggregated data as defined by the FAI framework focuses on two of the key dimensions: access and density. As with the prior research, each dimension is measured by three levels: Required, Suggested, and Innovative. If the study area meets the required criteria, it is awarded one point, and if not, it is deducted one point. Meeting the suggested or innovative requirements awards two or three points respectively, and do not subtract points if not present. For each dimension, the highest obtainable score is a 7, and the lowest is a −1. Scores for aggregated data are summarized based on these two key dimensions (density and access) as defined by the FAI scorecard are provided below.

### 4.1. Density Evaluation Findings

The Density dimension of the Food Abundance Index focuses on the proportion of nutritious, healthy food outlets to non-nutritious alternatives within the study area. For this analysis, we calculated a series of ratios that corresponded with the three levels of analysis in the Food Abundance Index. The factors, point assignments, and raw ratios are described in Table 1.

From the analysis, this region was awarded the lowest possible score for the Density dimension, earning a −1, as the ratio calculated between grocery stores and convenience stores was below the required threshold. In addition, there is a low quantity of organic food outlets and grocery stores in the communities in the Mid-Mon Valley region compared to fast food and convenience stores. The majority of food outlets in the Mid-Mon Valley are restaurants, as full service and fast-food restaurants account for 84.57% of food outlets. Excluding restaurants, there is a high concentration of convenience stores, corner stores, and food bank distribution centers in the area. This district also has a geographic challenge of food outlets being distributed on both sides of the Monongahela River, with limited bridge and transportation access between the district communities. This exacerbates the problem of a smaller amount of accessible healthier food outlets (e.g., limited assortment stores, supermarkets, grocery stores). While physical distance from these sparse stores may be minimal, the travel times can be extended due to heavy bridge traffic, congestion and lack of public transportation options.

### 4.2. Access Evaluation Findings

The Access dimension of the Food Abundance Index focuses on the ability to contact healthy, nutritious food with ease. The required level for this dimension is the presence of a mainstream grocery store within walking distance of a public transportation stop. The suggested level awards two points for the presence of a farmer’s market or organic store, and the innovative level requires members of the community to have access to community-based nutrition education and support. An analysis of the Mid-Mon Valley region reveals it has all these qualities, and thus earns six of six possible points. However, additional findings from the application reveal there may be underlying issues despite the positive scoring.

The latest Food Abundance Index application data (see Table 2) shows nine locations in this region lacking public transportation stops within 0.25 miles out of a total of 61 known food outlets (14.75%). In addition, seven of the census tracts in the district have an estimated less than 30% of households owning at least one car. For households with vehicles, auto congestion may inhibit access, as it is relatively high throughout this district. This high rate of congestion may also increase travel times for residents that have to take a bus to their primary food outlet. Food outlets reside on both sides of the river that splits this district, which poses a challenge to people traveling across bridges to access them. The sparsity of bridges, high congestion, and presence of food outlets without nearby transportation stops act as barriers for residents to acquire healthy food thus, reducing overall access. The findings from our data aggregation for the access dimension of the FAI are summarized in Table 2.

### 4.3. Key Findings from the FAI Analysis

Aggregated data for both access and density reveal that the Mid-Mon Valley is susceptible to experiencing significantly low levels these two dimensions of food security. Within our aggregated data we noted that there were several factors that contributed to the low scores for access and density. These contributing factors include lack of walkable public transportation to area food outlets, lack of vehicle ownership, and traffic congestion in the area. In addition, lack of availability of a diverse variety of local food outlets were also identified as critical factors contributing to low scores for both access and density dimensions. The aggregation across multiple existing data sources in our evaluation of the access and density dimensions also yielded specific targeted areas of focus that are highlighted as part of our findings (see Figure 1).

The presence of community-based food education programs and limited organic stores reveal potential and the need for further community-based development. While some outlets did not have adequate access to public transportation, this accounted for some of the outlets in the area. External data revealed other access challenges, such as congestion and lack of access may be defined by factors outside of the specific FAI scoring methodology and calculation. This further suggests the necessity for future qualitative community surveying as a complement to the FAI scorecard in order to address issues such as how individuals and families cope with lack of access, density and transportation resources that may not be apparent in the existing data sources used in the current calculations and scoring of the FAI dimensions (access and density).

## 5. Discussion

Our application of the enhanced FAI tool provides valuable insight into key dimensions of access and density that should be addressed for the community studied within the current research. Based on the findings from research, the Food Abundance Index provides a robust measurement tool to assess the extent of food insecurity within a community or neighborhood. A major contribution of this work is to present a new multi-dimensional measure of food insecurity that strengthens the link between the level of food security as both an ethical and social responsibility issue. We suggest that the adoption and utilization of data produced by the FAI offers a number of policy and management implications to help bring about sustainable social change in the food supply chain and address this as a wicked social problem [46].

The FAI can help communities understand that strategies like opening a mainstream grocery store in a food desert are just a starting point. Food security is a complex multi-dimensional issue and a host of strategies that address all aspects of food security will be needed to eliminate this social problem. These include working with stores to add healthy, nutritious foods to their inventory, working with businesses to market healthy foods, providing educational support for healthy consumption, incentivizing people to buy healthy foods and making alternate sources of healthy foods like farmers markets accessible to all, especially low-income populations by acceptance of food stamps at farmer markets [47]. It also suggests that business and workforce development efforts are needed to build capacity across all aspects of sustainability (economic, social and environmental) [48]. Resident’s food needs can also be used as an economic driver to undertake sustainable policies and market-based enterprises that allow equitable food access. Our unique scorecard approach used by the FAI to measure food security can also be useful to communities to track improvement in levels of food insecurity over time based on intervention strategies implemented over time.

An ethics and social responsibility perspective asserts that ‘life’ in community is a public good, which should benefit all and be available to all in society [49]. Food security is one such good that has been deemed important for the welfare of society. However, the present food system models of production, distribution and supply chain management can negatively impact power and minoritized communities. While the public sector has traditionally shouldered the responsibility for addressing issues related to hunger and food insecurity, the private sector if it does not provide for the marginalized and food insecure, is losing out on a whole market of value-demanding consumers [50].

Identifying food insecure areas is the first steps in helping communities reach the ultimate goal of building an ethical and equitable food system. The information, awareness and understanding of the current state of access to and availability of healthy, affordable foods gathered through tools such as the Food Abundance Index could provide the basis for change, action and future direction to academics and practitioners. Thus, our work in developing and distributing the FAI tool reflects our attempt to bridge the gap and expand the boundaries between academia and practitioners using robust methodology and innovative assessment tool.

Our tool has been designed to be accessible and easy to use by practitioners, especially those with limited financial and research resources to study the problems of food security, identify shortcoming in the local food system and also develop evidence-based solutions/policies. In addition, the FAI tool considers food security across multiple dimensions and provides a complete picture of food security in a community as related to key dimensions (access and diversity) examined in the current research and the additional dimensions of quality, density and affordability not measured in this research but provide opportunity for future research. As a measurement tool, the FAI can not only assess whether a food desert exists within a particular community, but as our current findings indicate, it can also be used to identify where gaps or problems may exist that may cause a breakdown in food security in the future. Lastly, the FAI uses a scorecard approach to measure for different levels of food security, track improvement in levels of food security over time and provide a benchmark that practitioners can use for addressing the issue of food security in other geographic areas.

We argue that the adoption and utilization of data produced by the FAI offers a number of potential contribution and benefits. Analysis of geographic areas using the FAI will help lay the foundation for reducing communities’ food insecurities and improving economic status, health, and overall well-being. We expect that use of the FAI to also stimulate additional academic research by providing researchers with a comprehensive tool to measure levels of food security as well as outcomes and the impact of change efforts. Lastly, we anticipate use of the FAI tool to facilitate collaboration, knowledge sharing and dialog between academic and community-based research based on the common language, methodology and focus provided by this new approach to understand the dimensions and impact of food insecurity as a key social issue.

## 6. Conclusions

Our analysis identified several issues and assets that can be utilized by business, community and public sector organizations to address the issue of access and density to develop both ethical and socially responsible solutions within this region. These issues and assets are summarized below as key implications of our work using the expanded FAI tool within this area examined in the current research.

### 6.1. Key Gaps and Issues Identified in the Mid-Mon Valley

Collapse of industrial era economic infrastructure.Legacy of limited attention to environmental sustainability in industrial practices.Gap in availability of and reasonable access to healthy affordable food.Diminished regional food infrastructure related to transformation.Lack of investment on the regional level in food systems infrastructure.Disproportionate representation of lower income, lower levels of business ownership and employment by the age and race/ethnicity demographic groups.Community negatively impacted by the COVID-19 global pandemic.Higher number of health disparities associated with poor nutrition and lack of healthy, affordable food options.

### 6.2. Key Assets and Opportunities Identified in the Mid-Mon Valley

A greater metro region marketplace of over 1.2 million people, with a total of 537,000 households, spending approximately $3 billion plus per year combined on food and beverage needs.A “foodshed” region with the soils, water, people and production capable of supplying an estimated potential of at least 25% ($750 million in sales) or more of the annual total food and beverage market demands of the greater metro area this district resides within.Greater Foodshed region contains over 7 million acres in production is already in the top ten agriculture producing regions in the US.A capable workforce with embedded historical experience in industrial production, that can be evolved and adapted to in region food production.An extensive business network of entrepreneurs tied to former and existing industrial and retail enterprises that could be resources to rebuild the local food economy.

## 7. Future Directions

While the use of aggregated data to examine the dimensions of the Food Abundance Index as a tool are important indicators, it should be viewed not only as a one-time snapshot but as a way of measuring continued progress and ongoing improvement. The use of the FAI tool as the framework for collecting and aggregating existing data can help to curate data in real time as an integral part of developing comprehensive policies, programs, and projects to address challenges and deficiencies. Our current work points to the future potential of the FAI tool to shape ongoing efforts to leverage new and existing data in a meaningful way that can provide direction and feedback to policymakers, community members and business leaders.

The use of existing data that is shaped and informed by the FAI framework can become an invaluable tool for developing a resilient and sustainable food economy both regionally and globally. Bringing together data from a multitude of sources is an important contribution of the current work for both research and practice. This tool can be beneficial in the future to move away from previous attempts to address food desert with single efforts solutions such as a single grocery or food outlet. The FAI framework together with comprehensive data sources can fill a critical need for evidence-based approaches that not only shape decision making but also measure its ongoing impact and effectiveness The further refinement of the FAI tool reflects our ongoing efforts to shape the conversation and efforts toward building a more resilient and sustainable food ecosystem that reduces disparities and moves us forward to a more equitable and just food ecosystem.

## Figures and Tables

**Figure 1 ijerph-19-10042-f001:**
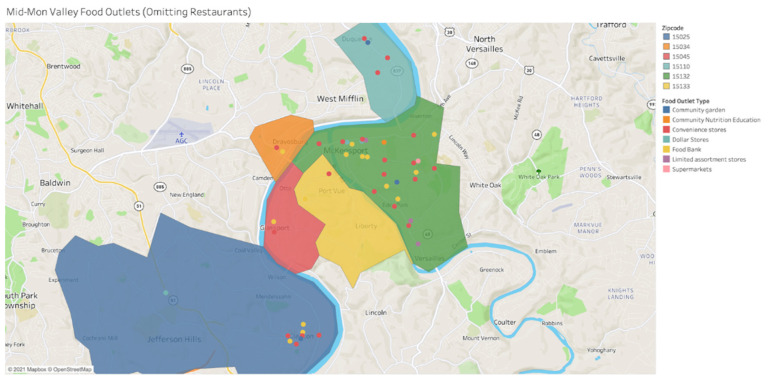
FAI Access to and Density Dimensions by Public Transportation Resources.

**Table 1 ijerph-19-10042-t001:** FAI Density Dimension Results.

Density Factor Criteria	FAI Points for Region	Mid-Mon Valley Ratio
Higher number of grocery stores to convenience stores	−1	1 grocery store to 7 convenience stores
Higher number of grocery stores and produce vendors to convenience stores	0	2 grocery stores/organic food outlets to 7 convenience stores
Higher number of local organic food source outlets to fast food and convenience stores	0	1 organic food outlet to 94 fast food/convenience stores

**Table 2 ijerph-19-10042-t002:** FAI Access Dimension Results.

Access Factor Requirement	FAI Points for Region
Presence of a mainstream grocery store within walking distance of a public transportation stop in the region	1
Presence of a farmer’s market or organic store in the region	2
Presence of community-based nutrition education support in the region	3

## Data Availability

The data presented in this study are available on request from the corresponding author.

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
