# Peer review of "Food Security as Ethics and Social Responsibility: An Application of the Food Abundance Index in an Urban Setting"

_ijerph, 2022, doi:10.3390/ijerph191610042_

Round 1
Reviewer 1 Report
The manuscript entitled ‘Food Security as Ethics and Social Responsibility: An Application of the Food Abundance Index in an Urban Settings’ is an interesting study and falls under the scope of the journal however, there is a scope for improvement. Kindly address the following points before the publication:
General comment:
There are several grammatical mistakes therefore, the MS must be thoroughly checked for English language and grammatical errors.
There are too many old references. Kindly replace them with new ones.
Specific comments:
Abstract:
It should be more meaningful and specific.
Introduction:
Kindly briefly describe the role of society as one of the pillars of sustainability in food security.
Line 40-42: Reframe the sentence.
Line 60-63: Rephrase the sentence and divide it into two.
Line 107: ‘society. and’? Remove the full stop.
Line 159: ‘measure and intensity’? What does it mean? Rectify the sentence.
Line 162: ‘and corporations. and reflect’? Rectify it.
Line 167: Either write ‘direct’ or ‘physical’. Rectify.
Line 202-210: It seems that this para has the repetitive information as given in the previous para (123-133). Therefore, merge it there.
3. Method: Kindly disclose the sample size and if a survey was carried out to obtain data, the questionnaire should be given as supplementary material.
Line 333: Provide a relevant reference to support the statement.
At the end, a separate section should be there as conclusions and future perspectives.
Author Response
Responses to feedback/suggestions are provided in the attached document.

Reviewer 2 Report
The manuscript reflects an interesting idea, a topical one, but it have be improved.
1. The abstract is not concludent enough. Maybe if you extend it will be better.
2. Check the accepted method of citation - with "[...]" - in author's guideline
3. The Introduction part have almost 4 pages, and the manuscript have 12 pages. It is extremely long!
4. The Methods are not clear described.
5. In my opinion this manuscript needs more data.
6. The link that you provided at the Supplementary Material is not available. This is the response:
" No Results Found. The page you requested could not be found. Try refining your search, or use the navigation above to locate the post."
7. The Author contribution have to be more specific (For research articles with several authors, the following statements should be used “Conceptualization, X.X. and Y.Y.; methodology, X.X.; software, X.X.; validation, X.X., Y.Y. and Z.Z.; formal analysis, X.X.; investigation, X.X.; resources, X.X.; data curation, X.X.; writing—original draft preparation, X.X.; writing—review and editing, X.X.; visualization, X.X.; supervision, X.X.; project administration, X.X.; funding acquisition, Y.Y. All authors have read and agreed to the published version of the manuscript.”)
8. Why are you thanking in Acknowledgements to the "Food21 of Pennsylvania"? Two of you, the authors, are affiliated to the "Food21 of Pennsylvania"....
9. The list of References is too short, and this is reflecting on the quality of manuscript. (see point no. 5). I am sure that you can resolve that.
The manuscript, in the current form, must be improved.
Author Response
Responses to reviewer comments/suggestions are provided in the attached document.

Round 2
Reviewer 2 Report
You have solved all the aspects that I have identified as needing to be corrected. Now the work can be published in its current form. Congratulations!